# CDCP1 Expression Is a Potential Biomarker of Poor Prognosis in Resected Stage I Non-Small-Cell Lung Cancer

**DOI:** 10.3390/jcm11020341

**Published:** 2022-01-11

**Authors:** Yunha Nam, Chang-Min Choi, Young Soo Park, HyunA Jung, Hee Sang Hwang, Jae Cheol Lee, Jung Wook Lee, Jung Eun Lee, Jung Hee Kang, Byung Hun Jung, Wonjun Ji

**Affiliations:** 1Asan Medical Center, Department of Pulmonary and Critical Care Medicine, University of Ulsan College of Medicine, Seoul 05505, Korea; siriusnyh@gmail.com (Y.N.); ccm@amc.seoul.kr (C.-M.C.); 2Asan Medical Center, Department of Oncology, University of Ulsan College of Medicine, Seoul 05505, Korea; jclee@amc.seoul.kr; 3Asan Medical Center, Department of Pathology, University of Ulsan College of Medicine, Seoul 05505, Korea; youngspark@amc.seoul.kr (Y.S.P.); eldersage@empas.com (H.S.H.); 4Asan Medical Center, Asan Institute for Life Sciences, University of Ulsan College of Medicine, Seoul 05505, Korea; hyuna0826@naver.com; 5Therapeutic Antibody R&D Center, Theranotics Co., Ltd., Seoul 05842, Korea; jwlee@theranotics.com (J.W.L.); jelee@theranotics.com (J.E.L.); jhkang@theranotics.com (J.H.K.); bhjung07@theranotics.com (B.H.J.)

**Keywords:** lung cancer, non-small-cell lung cancer, surgical resection, CDCP1, biomarker, prognosis

## Abstract

Background: Although early-stage lung cancer has increased owing to the introduction of screening programs, high recurrence rate remains a critical concern. We aimed to explore biomarkers related to the prognosis of surgically resected non-small-cell lung cancer (NSCLC). Methods: In this retrospective study, we collected medical records of patients with NSCLC and matched tissue microarray blocks from surgical specimens. Semiquantitative immunohistochemistry was performed for measuring the expression level of fibroblast activation protein-alpha (FAP-α), Jagged-1 (JAG1), and CUB-domain-containing protein 1 (CDCP1). Results: A total of 453 patients who underwent complete resection between January 2011 and February 2012 were enrolled; 55.2% patients had stage I NSCLC, and 31.1% presented squamous cell carcinoma. Disease stage was a significant risk factor for recurrence and death, and age ≥ 65 years and male sex were associated with poor overall survival. FAP-a and JaG1 were not related to survivals, while CDCP1-expressing patients exhibited poor disease-free and overall survival. Moreover, CDCP1 expression in stage I NSCLC was significantly associated with recurrence. Conclusions: Old age, male sex, and high pathological stage were poor prognostic factors in patients with NSCLC who underwent surgical resection. Furthermore, CDCP1 expression could serve as a biomarker for poor prognosis in stage I NSCLC.

## 1. Introduction

Lung cancer is the most common cause of cancer death worldwide [1]. Among all lung cancer cases, approximately 85% in Korea and 80% worldwide were confirmed to be non-small-cell lung cancer (NSCLC) [2]. These patients face a poor prognosis and low 5-year survival rate despite development of novel treatments [3,4]. Lung cancer screening using low-dose computed tomography in a high-risk group was recently reported to significantly lower the lung cancer mortality rate [5]; however, approximately 30% of patients are diagnosed at the unresectable stage despite screening [6]. In addition, 30% to 55% of patients who undergo curative resection for NSCLC develop recurrence and die of the disease [7]. Moreover, based on the TNM staging system, the 5-year survival rate reached 80–90% in stage I even in patients who underwent surgical resection but decreased to 60–70% at stage II, showing differences in prognosis even in lung cancer following surgical resection [8]. Therefore, identification of factors related to recurrence is critical for improving the prognosis of lung cancer.

Jagged-1 (JAG1), a cell surface protein which is one of five canonical ligands of Notch receptors, triggers Notch signaling through cell–cell interaction [9] and is associated with cancer angiogenesis, proliferation, and the epithelial-mesenchymal transition (EMT) signaling pathway [10,11]. *JAG1* acts as an oncogene in various cancers such as breast cancer, brain tumor, cervical cancer, colorectal cancer, and endometrial cancer [10]. It is additionally expressed in lung cancer, and it has been shown to promote cancer cell invasion and metastasis, suggesting that it may be clinically relevant [11].

CUB-domain-containing protein 1 (CDCP1) is a transmembrane protein overexpressed in breast, colon, and pancreatic cancer [12]. CDCP1 overexpression is known to activate several pathways which control cell adhesion [13]. In contrast, recent studies reported that loss of CDCP1 supports tumor cell proliferation by differentially regulating SRC activity in nonadherent conditions [14]. In addition, its overexpression was confirmed in lung cancer, and it could affect cancer progression by affecting the migration ability of lung cancer [15].

Fibroblast activation protein-alpha (FAP-α) is a type 2 transmembrane protein that is an important surface marker of cancer-associated fibroblasts that promotes cancer progression, cancer cell migration, invasion, and colony formation [16]. It also acts as an immune suppressor in the tumor microenvironment [17] and decreases survival in colon cancer [18] and hepatocellular carcinoma [19] patients. A recent in vitro study suggested that FAP-α facilitates proliferation of lung adenocarcinoma and that it could serve as a prognostic marker [20]. Although most cellular level studies on these biomarkers suggest such a possibility, prognosis due to overexpression is not well known.

Moreover, the current guideline recommends that only high-risk adjuvant chemotherapy be selectively performed according to the clinician’s decision among early-stage lung cancer patients who have previously undergone complete resection [21]. These high-risk patients are determined according to histopathologic or clinical features, and there are still not enough studies that establish prognostic factors related to recurrence other than in the early stage [22]. It is necessary and meaningful to find clinically viable biomarkers related to recurrence of early-stage lung cancer. Therefore, in this study, we aimed to explore prognostic biomarkers related to recurrence and death in NSCLC patients who underwent surgical resection.

## 2. Materials and Methods

### 2.1. Study Population

This study was a retrospective cohort study conducted using 504 tissue microarray (TMA) blocks collected from patients diagnosed with NSCLC who underwent complete surgical resection at Asan Medical Center between January 2011 and February 2012. Among the eligible patients, those who did not undergo immunohistochemistry for any of the markers (JAG1, CDCP1, or FAP-α) were excluded from our analysis (Appendix A). Clinicopathologic characteristics including survival data were retrospectively collected by a review of medical records. Tumors were staged according to the 7th edition of the American Joint Committee on Cancer tumor-node-metastasis staging system, and histologic grading and subtyping were performed in accordance with the World Health Organization’s guidelines. This study was approved by the institutional review board of Asan Medical Center (2020-0103, Seoul, Korea), and it conforms to the tenets of the Declaration of Helsinki.

### 2.2. TMA Production and Immunohistochemistry of Biomarkers

Tissue microarrays (TMAs) with 2 mm-diameter cores were constructed from representative tumor sections using formalin-fixed paraffin-embedded blocks. Immunohistochemical staining was performed on the TMA sections using the following antibodies: FAP-α (rabbit polyclonal, 1:600, Invitrogen, MA, USA, PA5-51057), CDCP1 (rabbit polyclonal, 1:100, Cell Signaling Technology, MA, USA, #4115), and JAG1 (rabbit monoclonal, 1:400, Abcam, Cambridge, UK). In brief, following deparaffinization and dehydration, heat-induced antigen retrieval was performed for 20 min in an antigen-retrieval buffer at pH 7.4 (for FAP-α), pH 7.5 (for CDCP1), or pH 7.2 (for JAG1) using a steam pressure cooker. The antigen–antibody reaction was detected using EnVision+ Dual Link System-HRP (Dako, Santa Clara, CA, USA) and visualized using 3,3′-diaminobenzidine (DAB+; Dako). Tissue sections were lightly counterstained with hematoxylin and examined by light microscopy. The positive controls were cell blocks prepared from HepG2 (FAFa), MDA-MB-231 (CDCP1), and HEK293 (JAG1). Isotype-matched control antibodies were used as negative controls.

The immunostaining evaluation was performed using a computer-assisted image analyzing software (Visiopharm version 4.5.1.324, Hoersholm, Denmark). Stained slides were scanned using NanoZoomer 2.0 HT (Hamamatsu Photonics, Hamamatsu City, Japan) at 20× objective magnification (0.5 μm resolution). Captured digital images were subsequently imported into the Visiopharm software. Each core was imported separately using the tissue microarray workflow of the program. For image analysis, first, the transformation of an image from one form to another (image processing) was performed for enhancing image structures of relevance for subsequent image segmentation. After training the system using digitally “painted” examples of the image, areas were defined in a process termed segmentation. The mean intensity of DAB in each defined image was used for the quantification of the expression. Immunoreactivity was assessed using the semiquantitative H-score (range 0–300), which was derived through the summation of each staining intensity (0–3) multiplied by the percentage (0–100) of immunoreactive cells with that intensity. For identifying the association of the immunoreactivity of possible metastatic markers with clinicopathological features and survival, when the average value of the quantified result of 2 cores was above 0.5 of staining intensity, the result was defined as positive immunoreactivity.

### 2.3. Statistical Analysis

All values are expressed as the median and interquartile range (IQR) for continuous variables or percentages for categorical variables. Student’s t-test or the Mann–Whitney U test was used for examining continuous data and the chi-square test or Fisher’s exact test for the categorical data.

Cox regression analysis was performed for identifying the factors affecting mortality, and subgroup analysis was performed for determining the overall survival and recurrence, depending on whether FAP-α, JAG1, and CDCP1 were overexpressed. Additionally, variables in the univariate analysis were entered into the multivariate models and we used multivariate analysis with backward selection. Survival analysis was performed using Kaplan–Meier analysis and log-rank test. For all statistical analysis, IBM SPSS statistics version 20 (IBM, Armonk, New York, NY, USA) was used; a statistically significant difference was defined as *p* value ≤ 0.05.

## 3. Results

### 3.1. Baseline Characteristics of Patients

A total of 453 patients with lung cancer who underwent surgical resection were included in this study. The median follow-up period was 87 months (IQR: 44–105 months). The baseline characteristics of the study population are presented in Table 1. The median age was 63 years (IQR: 57–50 years old); 309 patients (68.2%) were male and 64.7% were ever-smokers. According to the operation type, 85.4% of patients underwent lobectomy, 8.2% underwent sublobar resection, and 4.6% underwent bilobectomy. Regarding the stage at initial diagnosis, over half of the patients (55.2%) had stage I lung cancer and 41 patients (9.1%) had stage IIIA lung cancer. Lymph node metastasis was confirmed at the time of diagnosis in 144 patients (31.8%), among which 61 patients (13.5%) exhibited N1 disease and 82 patients (18.1%) exhibited N2 disease.

In total, 249 (55.0%) patients tested positive for JAG1; 265 (58.5%) tested positive for CDCP1, and 312 (68.9%) tested positive for FAP-α. JAG1 and CDCP1 were mainly highly expressed in the squamous cell carcinoma group, while FAP-α was highly expressed in the adenocarcinoma group. CDCP1 positivity was higher among men (*n* = 206, 66.7%) and smokers (*n* = 202, 68.9), and CDCP1 expression was significantly elevated in patients at stage II or higher (*p* < 0.001). The high expression of FAP-α in the adenocarcinoma group appears to be related to its high expression in nonsmoking women, a characteristic of the adenocarcinoma subtype.

### 3.2. Risk Factor Analysis of Recurrence within 5 Years after Surgery

Among the 453 patients, 130 (28.7%) patients exhibited recurrence within 5 years after surgery; the median duration was 15 months (IQR: 9–30 months) (Table 1). The 5-year disease-free survival (DFS) was 55.6%, and the DFS decreased significantly as the stage progressed, except at stage IV, at which the number of patients was too low (Figure 1). It showed the same trend, according to the subgroup analysis, for each biomarker. Only the stage was shown to affect the 5-year recurrence (*p* < 0.001), according to univariate and multivariate Cox analysis (Table 2). According to the analysis for the 5-year recurrence rate with respect to each biomarker, only the CDCP1-positive group showed a significantly higher rate of recurrence within 5 years after surgical resection (0.658 vs. 0.738, *p* = 0.039) (Figure 2).

### 3.3. Risk Factor Analysis of 5-Year Survival after Surgery

The 5-year overall survival (OS) of all patients was 69.3%, and the mortality rate showed an increasing trend according to the stage (*p* < 0.001). In addition, the subgroup analysis for each biomarkers showed that the mortality tended to increase in accordance with the stage (Table 1, Figure 1).

Univariate analysis showed that the following characteristics were associated with mortality: age greater than 65 years (odds ratio [OR], 2.086; 95% CI: 1.572–2.768; *p* < 0.001), male sex (OR, 1.724; 95% CI: 1.243–2.392; *p* = 0.001), ever-smoker (OR, 1.732; 95% CI: 1.264–2.372; *p* = 0.001), higher pathological stage at the time of diagnosis (*p* < 0.001), squamous cell type (OR, 1.564; 95 % CI: 1.175–2.082; *p* = 0.002), CDCP1 overexpression (OR, 1.498; 95 % CI: 1.118–2.007; *p* = 0.007) and previous sublobar resection (OR, 1.888; 95% CI: 1.219–2.923; *p* = 0.004). Multivariable Cox regression analysis showed that the following characteristics were associated with mortality: old age, (hazard ratio [HR], 1.813; 95% CI: 1.274–2.578; *p* = 0.001), male sex (HR, 1.542; 95 % CI: 1.024–2.323; *p* = 0.038), and higher initial stage (*p* < 0.001) (Table 3).

The expression of FAP-α or JAG1 did not indicate any difference in survival (median survival period: 105 months vs. not reached, *p* = 0.575, 106 months vs. not reached, *p* = 0.426), while the CDCP1-positive group exhibited a significantly poorer survival rate (median survival period: 109 months vs. not reached, *p* = 0.006) than the CDCP1-negative group (Figure 3).

### 3.4. Subgroup Analysis in Stage I Non-Small-Cell Lung Cancer

Subgroup analysis in stage I NSCLC was performed to evaluate the prognostic differences according to expression of CDCP1. Clinical information on stage I NSCLC patients is summarized in Appendix A. The characteristics were similar to those of the patients described above. However, the univariate and multivariate Cox proportional hazard model showed that CDCP1 positivity was significantly associated with recurrence within 5 years (HR, 1.967; 95 % CI: 1.104–3.504; *p* = 0.022) (Table 4); mortality was associated with old age (HR, 3.420; 95 % CI: 2.082–5.619; *p* < 0.001) and squamous cell carcinoma (HR, 1.907; 95 % CI: 1.181–3.081; *p*= 0.008) (Appendix A).

## 4. Discussion

We determined that age greater than 65 years, male sex, and high pathological stage were significant indicators of poor prognosis in terms of recurrence and death in patients with NSCLC who underwent surgical resection. Interestingly, CDCP1 expression was associated with poor DFS and OS; however, the association was not statistically significant according to Cox multivariate analysis. However, CDCP1 expression in stage I lung cancer was significantly associated with recurrence within 5 years after surgical resection.

Several studies have suggested factors associated with cancer recurrence after surgical resection [23,24]. Taylor et al. reported that preoperative tumor maximum standardized uptake value (SUV_max_) greater than 5, preoperative radiation, and pathological stage II and III disease could act as risk factors for recurrence [23]. An analysis of 373 patients undergoing potentially curative resection of NSCLC from 2000 through 2005 showed that lymphatic or vascular invasion, use of chemotherapy, and diabetes were associated with local invasion, and non-squamous cell histology, pneumonectomy, and advanced TNM stage could act as risk factors for distant metastasis [24]. Concordant with most previous reports, in this study, the pathological stage was a significant risk factor for recurrence after surgical resection. Furthermore, we determined that CDCP1 expression was a risk factor for recurrence in stage I NSCLC.

In terms of OS after surgery, we observed that age greater than 65 years, male sex, and advanced stage were significantly associated with poor survival after surgery; these factors are already known to be associated with mortality after surgical resection [24,25]. Varlotto et al. reported that age, history of myocardial infarction, performance of a pneumonectomy, squamous cell histology, lymphatic or vascular invasion, and the number of positive N1 lymph nodes were significantly associated with survival [24]. An analysis of 3363 patients who underwent resection for primary lung cancer showed that low pathological stage, female sex, young age, and no alcohol abuse were favorable characteristics for 1-year survival [25].

Interestingly, in this study, the expression patterns of the biomarkers were different. JAG1 and CDCP1 were highly expressed in patients with squamous cell carcinoma, whereas FAP-α was highly expressed in patients with adenocarcinoma in our analysis. JAG1, triggers Notch signaling through cell–cell interaction and promotes cancer progression in several cancer types, such as HCC, gastric carcinoma, glioma, breast cancer, ovarian carcinoma, prostate cancer, and colorectal cancer [9]. In a study of 90 patients undergoing NSCLC surgery, JAG1 expression did not correlate with OS [11], which is consistent with our result. However, whether JAG1 can be used as a prognostic biomarker in lung cancer is not well known because of insufficient evidence. FAP-α decreases survival in several cancer types [18,19]. A previous study on 59 patients who underwent complete resection with NSCLC showed that the higher staining grade of FAP-α may be related to survival [26]; however, we did not find any differences in survival between FAP-α-positive and FAP-α-negative groups.

Dysregulated CDCP1 expression is known to affect progression in several cancers [27]. Ikeda et al. [28] conducted a multivariate Cox regression analysis of 200 lung adenocarcinoma patients and reported that high CDCP1 expression acts as an independent prognostic factor for OS and DFS. Dagnino et al. [29] suggest that circulating serum levels of CDCP1 provide additional information on future lung cancer risk, especially in patients who had tobacco exposure. Although this study was different from our study in which they measured plasma CDCP1 levels, our results were similar in terms of the possibility of CDCP1 being an early diagnostic marker considering the high percentage of smoking patients in the CDCP1-positive group. In our study, CDCP1 expression was associated with recurrence and death in patients undergoing surgery for lung cancer. The CDCP1-positive group exhibited a significantly higher proportion of male patients (66.7%, *p* < 0.001), smokers (68.9%, *p* < 0.001), and patients at a stage higher than II (*p* < 0.001) than the CDCP1-negative group. In multivariate Cox regression analysis, we observed that CDCP1 positivity was a significant risk factor for recurrence within 5 years after surgery in stage I NSCLC, which suggested that CDCP1 expression might be a clinically meaningful prognostic factor that could guide the delivery of adjuvant chemotherapy in patients with stage IB NSCLC. Platinum-based adjuvant chemotherapy after surgery has been reported to improve OS and DFS in completely resected stage IB NSCLC patients who presented factors indicating poor prognosis [30]. In addition, the current clinical guideline recommends that postoperative chemotherapy can be considered in patients with high-risk, margin-negative stage IB disease [21]. According to our findings, CDCP1-positive patients might be considered as a high-risk group for recurrence after surgery and could be possible candidates for adjuvant chemotherapy; although additional research is warranted for investigating this suggestion.

In this study, we observed that old age, male sex, and high pathological stage were significant indicators of poor prognosis in patients with NSCLC who underwent surgical resection. In addition, CDCP1 expression could serve as a biomarker of poor prognosis in terms of recurrence in stage I NSCLC. However, this study is a retrospective single-center study, and the possibility that sufficient numbers and statistical power could not be secured by using only patients who operated for a certain period for which TMA block production is possible is a limitation. Nevertheless, it is considered a meaningful study to confirm a significant difference in prognosis in the CDCP1 positive group. In the future, to verify this result, a prospective study including a sufficient number of patients will be conducted to determine a more accurate clinical significance. In addition, further studies are needed to determine whether CDCP1 can help to improve the actual prognosis by applying it to determine treatment strategies such as adjuvant chemotherapy and for prediction of recurrence in patients who have undergone complete resection in actual clinical practice.

## Figures and Tables

**Figure 1 jcm-11-00341-f001:**
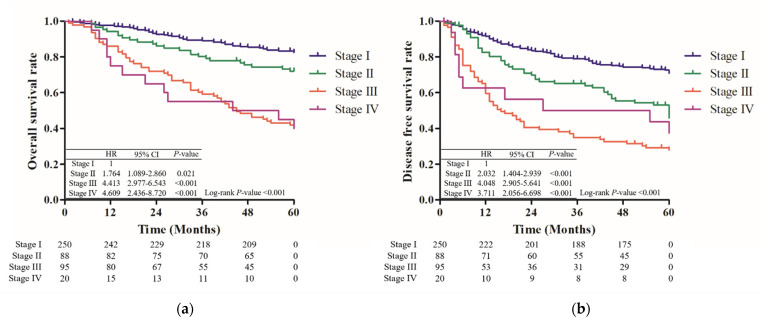
Overall survival and disease-free survival according to pathological stage. (**a**) Five-year overall survival according to pathologic stage. (**b**) Five-year disease-free survival according to pathological stage.

**Figure 2 jcm-11-00341-f002:**
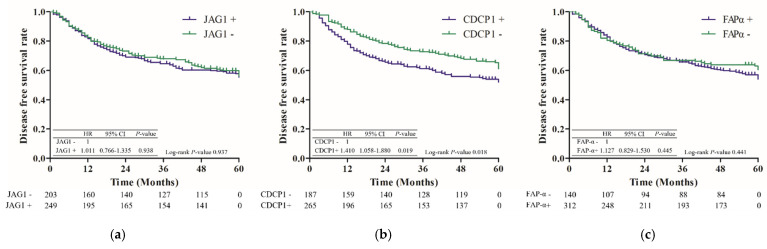
Disease free survival after surgery according to biomarker expression. (**a**) JAG1; (**b**) CDCP1; (**c**) FAP-α.

**Figure 3 jcm-11-00341-f003:**
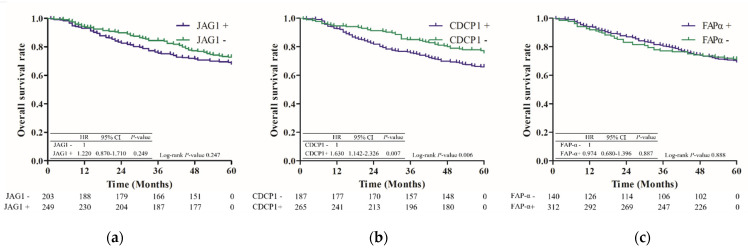
Five-year overall survival according to biomarker expression. (**a**) JAG1; (**b**) CDCP1; (**c**) FAP-α.

**Table 1 jcm-11-00341-t001:** Baseline characteristics over a median follow-up period of 87 months (IQR 44–105).

	All (*n* = 453)	JAG1 (+) (*n* = 249)	*p* Value	CDCP1 (+) (*n* = 265)	*p* Value	FAP-α (+) (*n* = 312)	*p* Value
Age, years, median (range)	63 (57–70)	63 (57–70)		63 (57–71)		63 (57–70)	
Sex, male, *n* (%)	309 (68.2)	177 (57.3)	0.168	206 (66.7)	<0.001	197 (64.0)	0.001
Ever-smokers, *n* (%)	293 (64.7)	167 (57.0)	0.268	202 (68.9)	<0.001	185 (63.6)	<0.001
Operation type			0.908		0.001		0.002
Lobectomy, *n* (%)	387 (85.4)	211 (54.7)		213 (55.2)		273 (70.7)	
Sublobar resection, *n* (%)	37 (8.2)	21 (56.8)		25 (67.6)		28 (75.7)	
Bilobectomy. *n* (%)	21 (4.6)	13 (61.9)		20 (95.2)		8 (38.1)	
Pneumonectomy, *n* (%)	8 (1.8)	4 (50.0)		7 (87.5)		3 (37.5)	
Adjuvant Chemotherapy, *n* (%)	21 (4.6)	9 (42.9)	0.494	10 (47.6)	0.568	16 (76.2)	0.745
Stage			0.602		<0.001		0.337
I, *n* (%)	250 (55.2)	143 (57.4)		125 (50.2)		178 (71.2)	
II, *n* (%)	88 (19.4)	48 (54.5)		66 (75.0)		55 (63.2)	
III, *n* (%)	95 (21.0)	49 (51.6)		60 (63.2)		63 (66.3)	
IV, *n* (%)	20 (4.4)	9 (45.0)		14 (70.0)		16 (80.00	
Histology			0.035		<0.001		<0.001
Squamous cell, *n* (%)	141 (31.1)	88 (62.4)		123 (87.2)		64 (45.7)	
Non-Squamous, *n* (%)	312 (68.9)	161 (51.8)		142 (45.7)		248 (79.5)	
Node meta			0.813		0.514		0.988
N0, *n* (%)	306 (67.5)	173 (56.5)		169 (55.2)		210 (68.6)	
N1, *n* (%)	61 (13.5)	35 (57.4)		44 (72.1)		39 (63.9)	
N2, *n* (%)	82 (18.1)	38 (46.3)		49 (59.8)		60 (73.2)	
N3, *n* (%)	1 (0.2)	1 (100.0)		1 (100.0)		1 (100.0)	
Five-year disease-free survival	130 (28.7)	72 (28.9)	0.009	84 (31.7)	0.006	96 (30.8)	<0.001
Stage I, *n* (%)	50 (20.0)	33 (23.1)		32 (25.6)		39 (21.9)	
Stage II, *n* (%)	28 (31.8)	15 (31.3)		19 (28.8)		17 (30.9)	
Stage III, *n* (%)	48 (50.5)	23 (46.9)		30 (50.0)		36 (57.1)	
Stage IV, *n* (%)	4 (20.0)	1 (11.1)		3 (21.4)		4 (25.0)	
Duration, months, median (range)	15 (9–30)	15 (9–30)	0.708	13 (6–26]	0.289	15 (8–31)	0.496
Five-year overall survival	139 (30.7)	82 (32.9)	<0.001	94 (35.5)	<0.001	96 (30.8)	<0.001
Stage I, *n* (%)	45 (18.0)	28 (19.6)		30 (24.0)		31 (17.4)	
Stage II, *n* (%)	26 (29.5)	15 (31.3)		20 (30.3)		16 (29.1)	
Stage III, *n* (%)	56 (58.9)	33 (67.3)		36 (60.0)		41 (65.1)	
Stage IV, *n* (%)	12 (60.0)	6 (66.7)		8 (57.1)		8 (50.0)	

Others: 25 patients underwent wedge resection; 12 patients underwent segmentectomy.

**Table 2 jcm-11-00341-t002:** Cox regression analysis for recurrence within 5 years after surgery.

	Univariate Analysis		Multivariate Analysis	
Parameter	HR (95 % CI)	*p* Value	HR (95 % CI)	*p* Value
Age (≥65)	1.018 (0.720–1.439)	0.921	0.928 (0.648–1.329)	0.683
Male	1.192 (0.817–1.740)	0.363	1.020 (0.517–2.011)	0.955
Ever-smoker	1.238 (0.857–1.788)	0.255	1.041 (0.529–2.047)	0.907
Operation type		0.745		0.443
Lobectomy	1		1	
Sublobar resection	1.192 (0.641–2.217)	0.578	1.504 (0.661–3.421)	0.330
Bilobectomy	1.360 (0.633–2.920)	0.431	0.792 (0.237–2.654)	0.706
Pneumonectomy	1.470 (0.467–4.629)	0.511	1.081 (0.379–3.087)	0.884
Stage		<0.001		<0.001
I	1		1	
II	1.781 (1.121–2.829)	0.015	1.769 (1.080–2.900)	0.024
III	3.905 (2.621–5.818)	<0.001	4.165 (2.752–6.303)	<0.001
IV	1.335 (0.482–3.698)	0.578	1.081 (0.379–3.087)	0.884
Histology				
Adenocarcinoma	0.835 (0.583–1.196)	0.325	0.859 (0.310–2.377)	0.769
Squamous cell	1.159 (0.802–1.673)	0.433	0.796 (0.265–2.397)	0.686
Biomarker				
JAG1	1.025 (0.726–1.449)	0.887	1.037 (0.725–1.483)	0.842
CDCP1	1.456 (1.016–2.086)	0.041	1.282 (0.859–1.913)	0.225
FAP-α	1.313 (0.884–1.951)	0.177	1.489 (0.960–2.310)	0.076
Adjuvant Chemotherapy	0.829 (0.365–1.880)	0.653	0.830 (0.361–1.909)	0.661

Abbreviations; JAG1: Jagged-1, CDCP1: CUB-domain-containing protein 1, FAP-α: Fibroblast activation protein-alpha.

**Table 3 jcm-11-00341-t003:** Cox regression analysis for 5-year overall survival.

	Univariate Analysis		Multivariate Analysis	
Parameter	HR (95 % CI)	*p* Value	HR (95 % CI)	*p* Value
Age (≥65)	2.086 (1.572–2.768)	<0.001	1.813 (1.274–2.578)	0.001
Male	1.724 (1.243–2.392)	0.001	1.542 (1.024–2.323)	0.038
Ever-smoker	1.732 (1.264–2.372)	0.001	0.934 (0.500–1.747)	0.832
Operation type		0.029		0.064
Lobectomy	1		1	
Sublobar resection	1.888 (1.219–2.923)	0.004	0.946 (0.424–2.110)	0.892
Bilobectomy	1.151 (0.608–2.179)	0.667	0.435 (0.103–1.831)	0.256
Pneumonectomy	0.561 (0.139–2.261)	0.416	2.002 (1.148–3.490)	0.014
Stage		<0.001		<0.001
I	1		1	
II	1.864 (1.281–2.712)	0.001	1.677 (1.006–2.795)	0.047
III	3.612 (2.592–5.031)	<0.001	4.961 (3.269–7.530)	<0.001
IV	3.556 (2.008–6.297)	<0.001	3.782 (1.919–7.455)	<0.001
Histology				
Adenocarcinoma	0.624 (0.471–0.828)	0.001	0.809 (0.293–2.237)	0.683
Squamous cell	1.564 (1.175–2.082)	0.002	1.024 (0.342–3.068)	0.966
Biomarker				
JAG1	1.121 (0.846–1.485)	0.427	1.315 (0.930–1.861)	0.122
CDCP1	1.498 (1.118–2.007)	0.007	1.170 (0.784–1.745)	0.443
FAP-α	1.092 (0.804–1.483)	0.573	1.103 (0.730–1.666)	0.640
Adjuvant Chemotherapy	0.966 (0.511–1.825)	0.915	0.855 (0.372–1.963)	0.712

Abbreviations; JAG1: Jagged-1, CDCP1: CUB-domain-containing protein 1, FAP-α: fibroblast activation protein-alpha.

**Table 4 jcm-11-00341-t004:** Cox regression analysis for recurrence within 5 years after surgery in stage I non-small-cell lung cancer.

	Univariate Analysis		Multivariate Analysis	
Parameter	HR (95 % CI)	*p* Value	HR (95 % CI)	*p* Value
Age (≥65)	1.239 (0.711–2.162)	0.45	1.295 (0.737–2.273)	0.368
Male	1.654 (0.892–3.067)	0.11	1.449 (0.774–2.714)	0.247
Ever-smoker	1.490 (0.830–2.675)	0.182	0.864 (0.298–2.503)	0.788
Operation type		0.265		0.624
Lobectomy	1		1	
Sublobar resection	1.799 (0.806–4.016)	0.152	1.194 (0.337–4.232)	0.784
Bilobectomy	1.731 (0.536–5.597)	0.359	1.516 (0.642–3.579)	0.342
Histology				
Adenocarcinoma	0.580 (0.320–1.051)	0.073	0.913 (0.119–7.035)	0.93
Squamous cell	1.710 (0.934–3.132)	0.082	1.299 (0.632–2.670)	0.476
Biomarker				
Jagged-1	1.443 (0.804–2.591)	0.219	1.353 (0.745–2.455)	0.321
CDCP-1	1.967 (1.104–3.504)	0.022	1.967 (1.104–3.504)	0.022
FAP-α	1.436 (0.736–2.805)	0.289	1.430 (0.729–2.807)	0.299
Adjuvant Chemotherapy	0.801 (0.195–3.295)	0.759	0.820 (0.194–3.463)	0.788

Abbreviations; JAG1: Jagged-1, CDCP1: CUB-domain-containing protein 1, FAP-α: fibroblast activation protein-alpha.

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
