# Peer review of "CDCP1 Expression Is a Potential Biomarker of Poor Prognosis in Resected Stage I Non-Small-Cell Lung Cancer"

_jcm, 2022, doi:10.3390/jcm11020341_

Round 1

Reviewer 1 Report

In this article, Nam et. al. explored CDCP1 expression for its prognostic potential in resected lung cancer. The manuscript requires improvements. Here are my brief comments: -

  1. What does should thoroughly check the manuscript for English structures.
  2. For example: However, although most studies on these biomarkers” should be corrected for English.
  3. Authors can improve KM plots. Log-rank p values and hazard ratios with 95% C.I should be included.
  4. In table 2, multivariate analysis, authors can run a separate analysis in which other covariates are also included. If authors don’t want to include it in the main text, they can append it as supplementary information.
  5. In tables, parameter columns, authors can mention the number of patients. For example, the readers should get the information of (n) of each parameter.
  6. A separate segment can be included to discuss the functional/biological relevance of these molecules. A table and figure describing the same can be ended to provide a perspective to the readers.

Author Response

In this article, Nam et. al. explored CDCP1 expression for its prognostic potential in resected lung cancer. The manuscript requires improvements. Here are my brief comments

Q1. What does should thoroughly check the manuscript for English structures.

For example: However, although most studies on these biomarkers” should be corrected for English.

Author’s response: The revision was done by an established editing company. The edit was done by a native English-speaking medical editor. Certification for the correction is attached.    

Q2. Authors can improve KM plots. Log-rank p values and hazard ratios with 95% C.I should be included.

Author’s response: Thank you for your valuable feedback. We have now added the information of Log-rank p value and hazard ratios with 95% C.I

Q3. In table 2, multivariate analysis, authors can run a separate analysis in which other covariates are also included. If authors don’t want to include it in the main text, they can append it as supplementary information.

Author’s response: Thank you for pointing out this important aspect. Based on your suggestion, we performed an additional multivariate analysis with covariates of lymph node metastasis and we have added it to the Table 2.

Table 2. Cox regression analysis for recurrence within 5 years after surgery

Univariate analysis

Multivariate analysis

Parameter

HR (95 % CI)

P value

HR (95 % CI)

P value

Age (≥65)

1.018 (0.720–1.439)

0.921

Male

1.192 (0.817–1.740)

0.363

Ever-smoker

1.238 (0.857–1.788)

0.255

Operation type

0.745

Lobectomy

1

Sublobar resection

1.192 (0.641–2.217)

0.578

Bilobectomy

1.360 (0.633–2.920)

0.431

Pneumonectomy

1.470 (0.467–4.629)

0.511

Node meta

<0.001

N0

1

N1

1.184 (0.690–2.032)

0.540

N2

2.766 (1.875–4.082)

<0.001

Stage

<0.001

<0.001

I

1

1

II

1.781 (1.121–2.82+)

0.015

1.781 (1.121–2.82+)

0.015

III

3.905 (2.621–5.818)

<0.001

3.905 (2.621–5.818)

<0.001

IV

1.335 (0.482–3.698)

0.578

1.335 (0.482–3.698)

0.578

Histology

Adenocarcinoma

0.835 (0.583–1.196)

0.325

Squamous cell

1.159 (0.802–1.673)

0.433

Biomarker

JAG1

1.025 (0.726–1.449)

0.887

CDCP1

1.456 (1.016–2.086)

0.041

FAP-α

1.313 (0.884–1.951)

0.177

Adjuvant Chemotherapy

0.829 (0.365–1.880)

0.653

Abbreviations ; JAG1: Jagged-1, CDCP1: CUB-domain-containing protein 1, FAP- α: Fibroblast activation protein-alpha

Q4. In tables, parameter columns, authors can mention the number of patients. For example, the readers should get the information of (n) of each parameter.

Author’s response: We added the information that you mentioned in the Tables.   

Table

Q5. A separate segment can be included to discuss the functional/biological relevance of these molecules. A table and figure describing the same can be ended to provide a perspective to the readers.

Author’s response: Thank you for your valuable feedback. Several details of oncogenic and prognostic roles for each molecule have been added to the introduction section. However, figures or tables were not added because it would deviate the focus from the results of our study.

Introduction section (Page 5, line 103-106)

Jagged-1 (JAG1), a cell surface protein which is one of five canonical ligands of Notch receptors, triggers Notch signaling through cell–cell interaction [9] and is associated with cancer angiogenesis, proliferation, and the epithelial-mesenchymal transition (EMT) signaling pathway [10,11]. JAG1 acts as an oncogene in various cancers such as breast cancer, brain tumor, cervical cancer, colorectal cancer, and endometrial cancer [10]. It is additionally expressed in lung cancer, and it has been shown to promote cancer cell invasion and metastasis, suggesting that it may be clinically relevant [11].

Introduction section (Page 5, line 110-114)

CUB-domain-containing protein 1 (CDCP1) is a transmembrane protein overexpressed in breast, colon, and pancreatic cancer [12]. CDCP1 overexpression is known to activate several pathways which control cell adhesion [13]. In contrast, recent studies reported that loss of CDCP1 supports tumor cell proliferation by differentially regulating SRC activity in nonadherent conditions [14].In addition, its overexpression was confirmed in lung cancer, and it could possibly affect cancer progression by affecting the migration ability of lung cancer [15].

Introduction section (Page 6, line 117-121)

Fibroblast activation protein-alpha (FAP-α) is a type 2 transmembrane protein that is an important surface marker of cancer-associated fibroblasts that promotes cancer progression, cancer cell migration, invasion, and colony formation [16]. While it also acts as an immune suppressor in the tumor microenvironment [17], and is known to decrease survival in colon cancer [18], and hepatocellular carcinoma [19] patients. A recent in vitro study suggested that FAP-α facilitates the proliferation of lung adenocarcinoma and that it could serve as a prognostic marker [20]. Although most cellular level studies on these biomarkers suggest such a possibility, prognosis due to overexpression is not well known.

Discussion section (Page 13, line 289-295)

Interestingly, in this study, the expression patterns were different with respect to biomarkers. JAG1 and CDCP1 were highly expressed in patients with squamous cell carcinoma, whereas FAP-α was highly expressed in patients with adenocarcinoma in our analysis. JAG1, triggers Notch signaling through cell–cell interaction and promotes cancer progression in several cancer types, such as HCC, gastric carcinoma, glioma, breast cancer, ovarian carcinoma, prostate cancer, and colorectal cancer [9]. In a study on 90 patients undergoing NSCLC surgery, JAG1 expression did not correlate with OS [11], which is consistent with our result. However, whether JAG1 can be used as a prognostic biomarker in lung cancer is not well known because of insufficient evidence. FAP-α decreases survival in several cancer types [18,19]. A previous study on 59 patients who underwent complete resection with NSCLC showed that the higher staining grade of FAP-α may be related to survival [26]; however, we did not find any differences in survival between FAP-α-positive and FAP-α-negative groups.

Discussion section (Page 13, line 303-308)

Dysregulated CDCP1 expression is known as acts on cancer progression [27]. Ikeda et al.[28] conducted a multivariate Cox regression analysis of 200 lung adenocarcinoma patients and reported that high CDCP1 expression acts as an independent prognostic factor for OS and DFS. Dagnino et al.[29] suggest that circulating serum levels of CDCP1 provide additional information on future lung cancer risk, especially in patients who had tobacco exposure. Although this study was different from our study in which they measured plasma CDCP1 levels, our results were similar in terms of the possibility of CDCP1 being an early diagnostic marker considering the high percentage of smoking patients in the CDCP1-positive group. In our study, CDCP1 expression was associated with recurrence and death in patients undergoing surgery for lung cancer. The CDCP1-positive group exhibited a significantly higher proportion of male patients (66.7 %, p<0.001), smokers (68.9 %, p<0.001), and patients at a stage higher than II (p<0.001) than the CDCP1-negative group.

Reviewer 2 Report

The most common cancer in both men and women is lung cancer. This malignancy has two types including SCLC and NSCLC that NSCLC is the most well-known one and is responsible for 85% of cases and death. Therefore, it is of interest that current study has focused on NSCLC. Another one is the novelty of current work and there is no similar experiment in this case. More importantly, it has focused on patients that is more directed towards clinic. Overall, current work is well. However, some issues should be addressed to improve quality of current work. Is running title necessary for this journal? There are only three references from 2020 and no study from 2021. I suggest authors to improve quality and visibility of their work by citing new articles and their discussion. The introduction section is a little concise and needs to be extended. The same is applied for conclusion section that should be elaborated and improved by adding limitations of current work and directing future studies. The first part of introduction is about lung cancer and it is not complete. I think it is better to explain more about subtypes of lung cancer. Suggested articles could be Doi, 10.1016/j.cellsig.2020.109871 and Doi, 10.1016/j.lfs.2021.119649. Is it possible to add a schematic figure about your work? Just a suggestion to improve quality of work. Some of the statements do not have reference and I suggest authors to cite relevant works. Is your sample size appropriate or it needs to be higher? If yes, please add in conclusion section that future studies should focus on more population. Before the conclusion, you have mentioned limitations of your work. Maybe a brief description in conclusion would be enough.

Author Response

The most common cancer in both men and women is lung cancer. This malignancy has two types including SCLC and NSCLC that NSCLC is the most well-known one and is responsible for 85% of cases and death. Therefore, it is of interest that the current study has focused on NSCLC. Another one is the novelty of current work and there is no similar experiment in this case. More importantly, it has focused on patients that are more directed towards the clinic. Overall, current work is well. However, some issues should be addressed to improve the quality of current work. Is running title necessary for this journal?

Author’s response: We have removed the running title.

There are only three references from 2020 and no study from 2021. I suggest authors improve the quality and visibility of their work by citing new articles and their discussions.

Author’s response: Thank you for pointing out this important aspect. We replaced some references with recently published ones. However, we had to limit ourselves to follow the reference limit set by the Journal of Clinical Medicine.

The introduction section is a little concise and needs to be extended. The same is applied to the conclusion section that should be elaborated and improved by adding limitations of current work and directing future studies.

Author’s response: We have added some information regarding the rationale of this study in the introduction section and limitation and directing of further studies in the Discussion section.

Introduction section (Page 5, line 92-93)

Lung cancer is the most common cause of death from cancer worldwide [1]. Among all lung cancer cases, approximately 85 % in Korea and 80 % worldwide were confirmed to be non-small cell lung cancer (NSCLC) [2], NSCLC patients suffer from poor prognosis and low 5-year survival rate despite the development of novel treatments [3,4]. Lung cancer screening using low-dose computed tomography in a high-risk group was recently reported to significantly lower the lung cancer mortality rate [5]; however, approximately 30 % of patients are diagnosed at the unresectable stage despite screening [6]. In addition, 30 to 55 % of patients who undergo curative resection for NSCLC develop recurrence and die of their disease [7].

Introduction section (Page 6, line 126-132)

Moreover, the current guideline recommends that only high-risk adjuvant chemotherapy be selectively performed according to the clinician's decision among early-stage lung cancer patients who have previously undergone complete resection [21]. These high-risk patients are determined according to histopathologic or clinical features, and there are still not enough studies that establish prognostic factors related to recurrence other than in the early stage [22]. It is necessary and meaningful to find clinically viable biomarkers related to the recurrence of early-stage lung cancer. Therefore, in this study, we aimed to explore prognostic biomarkers related to recurrence and death in NSCLC patients who underwent surgical resection.

Discussion section (Page 14, line 323-335)

In this study, we observed that old age, male sex, and high pathological stage were significant indicators of poor prognosis in patients with NSCLC who underwent surgical resection. In addition, CDCP1 expression could serve as a biomarker of poor prognosis in terms of recurrence in stage I NSCLC. However, this study is a retrospective single-center study, and the possibility that sufficient numbers and statistical power could not be secured by using only patients who operated for a certain period of time for which TMA block production is possible is a limitation. Nevertheless, it is considered a meaningful study to confirm a significant difference in prognosis in the CDCP1 positive group. In the future, to verify this result, a prospective study including a sufficient number of patients will be conducted to determine a more accurate clinical significance. In addition, further studies are needed to determine whether CDCP1 can help to improve the actual prognosis by applying it to determine treatment strategies such as adjuvant chemotherapy and for prediction of recurrence in patients who have undergone complete resection in actual clinical practice.

The first part of the introduction is about lung cancer and it is not complete. I think it is better to explain more about subtypes of lung cancer. Suggested articles could be Doi, 10.1016/j.cellsig.2020.109871 and Doi, 10.1016/j.lfs.2021.119649.

Author’s response: Thank you for your kind suggestion. We revised the introduction section to reflect the format of the articles you have recommended.

Introduction section (Page 5, line 103-106)

Jagged-1 (JAG1), a cell surface protein which is one of five canonical ligands of Notch receptors, triggers Notch signaling through cell–cell interaction [9] and is associated with cancer angiogenesis, proliferation, and the epithelial-mesenchymal transition (EMT) signaling pathway [10,11]. JAG1 acts as an oncogene in various cancers such as breast cancer, brain tumor, cervical cancer, colorectal cancer, and endometrial cancer [10]. It is additionally expressed in lung cancer, and it has been shown to promote cancer cell invasion and metastasis, suggesting that it may be clinically relevant [11].

Introduction section (Page 5, line 110-114)

CUB-domain-containing protein 1 (CDCP1) is a transmembrane protein overexpressed in breast, colon, and pancreatic cancer [12]. CDCP1 overexpression is known to activate several pathways which control cell adhesion [13]. In contrast, recent studies reported that loss of CDCP1 supports tumor cell proliferation by differentially regulating SRC activity in nonadherent conditions [14]. In addition, its overexpression was confirmed in lung cancer, and it could possibly affect cancer progression by affecting the migration ability of lung cancer [15].

Introduction section (Page 6, line 117-121)

Fibroblast activation protein-alpha (FAP-α) is a type 2 transmembrane protein that is an important surface marker of cancer-associated fibroblasts that promotes cancer progression, cancer cell migration, invasion, and colony formation [16]. While it also acts as an immune suppressor in the tumor microenvironment [17], and is known to decrease survival in colon cancer [18], and hepatocellular carcinoma [19] patients. A recent in vitro study suggested that FAP-α facilitates proliferation of lung adenocarcinoma and that it could serve as a prognostic marker [20]. Although most cellular level studies on these biomarkers suggest such a possibility, prognosis due to overexpression is not well known.

Is it possible to add a schematic figure about your work? Just a suggestion to improve the quality of work.

Author’s response: We have now added the schematic figure as supplementary materials

Some of the statements do not have references and I suggest authors cite relevant works.

Is your sample size appropriate or it needs to be higher? If yes, please add in the conclusion section that future studies should focus on more population. Before the conclusion, you have mentioned the limitations of your work. Maybe a brief description, in conclusion, would be enough.

Author’s response: Thank you for your suggestion. A summary of the limitations of the study has been added to the discussion section.

Discussion section (Page 14, line 323-335)

In this study, we observed that old age, male sex, and high pathological stage were significant indicators of poor prognosis in patients with NSCLC who underwent surgical resection. In addition, CDCP1 expression could serve as a biomarker of poor prognosis in terms of recurrence in stage I NSCLC. However, this study is a retrospective single center study, and the possibility that sufficient numbers and statistical power could not be secured by using only patients who operated for a certain period of time for which TMA block production is possible is a limitation. Nevertheless, it is considered a meaningful study to confirm a significant difference in prognosis in the CDCP1 positive group. In the future, to verify this result, a prospective study including a sufficient number of patients will be conducted to determine a more accurate clinical significance. In addition, further studies are needed to determine whether CDCP1 can help to improve the actual prognosis by applying it to determine treatment strategies such as adjuvant chemotherapy and for prediction of recurrence in patients who have undergone complete resection in actual clinical practice.

Round 2

Reviewer 1 Report

The authors have improved the manuscript. However, authors can add other variables in Multivariate analysis. For example in Table 2: age, male, etc are not excluded. Similarly, in Table 3 and Table 4. Authors have to include multiple covariates for the analysis to be multivariate. Authors need to explain its exclusion or include additional variables.

Author Response

Author’s response: Thank you for your kind supplementary explanation. In our multivariate analysis, we used backward elimination for variables with p<0.1 in univariate analysis. However, as you pointed out, it is considered appropriate to include important clinical variables such as age and sex in multivariate analysis. So, according to your opinions, we performed backward elimination for multivariate analysis include the relevant variables, and the results are updated in Table 2 to 4 and supplementary table S2. Also, the corresponding part of methods and results section were revised.

Methods section (Page 9, line 187-189)

Cox regression analysis was performed for identifying the factors affecting mortality, and subgroup analysis was performed for determining the overall survival and recurrence, depending on whether FAP-α, JAG1, and CDCP1 were overexpressed. Additionally, variables in the univariate analysis were entered into the multivariate models and we used multivariate analysis with backward selection. Survival analysis was performed using Kaplan–Meier analysis and log-rank test. For all statistical analysis, IBM SPSS statistics version 20 (IBM, Armonk, New York, USA) was used; a statistically significant difference was defined as p value ≤ 0.05.

Results section (Page 11, line 237-239)

Multivariable Cox regression analysis showed that the following characteristics were associated with mortality: old age, (hazard ratio [HR], 1.813; 95 % CI: 1.274–2.578; p=0.001), male sex (HR, 1.542; 95 % CI: 1.024–2.323; p=0.038), and higher initial stage (p<0.001) (Table 3).

Results section (Page 11, line 252-254)

However, the univariate and multivariate Cox proportional hazard model showed that CDCP1 positivity was significantly associated with recurrence within 5 years (HR, 1.967; 95 % CI: 1.104–3.504; p=0.022) (Table 4); mortality was associated with old age (HR, 3.420; 95 % CI: 2.082–5.619; p<0.001) and squamous cell carcinoma (HR, 1.907; 95 % CI: 1.181–3.081; p=0.008) (Table S2).

Table 2. Cox regression analysis for recurrence within 5 years after surgery

Univariate analysis

Multivariate analysis

Parameter

HR (95 % CI)

P value

HR (95 % CI)

P value

Age (≥65)

1.018 (0.720–1.439)

0.921

0.928 (0.648–1.329)

0.683

Male

1.192 (0.817–1.740)

0.363

1.020 (0.517–2.011)

0.955

Ever-smoker

1.238 (0.857–1.788)

0.255

1.041 (0.529–2.047)

0.907

Operation type

0.745

0.443

Lobectomy

1

1

Sublobar resection

1.192 (0.641–2.217)

0.578

1.504 (0.661–3.421)

0.330

Bilobectomy

1.360 (0.633–2.920)

0.431

0.792 (0.237–2.654)

0.706

Pneumonectomy

1.470 (0.467–4.629)

0.511

1.081 (0.379–3.087)

0.884

Stage

<0.001

<0.001

I

1

1

II

1.781 (1.121–2.829)

0.015

1.769 (1.080–2.900)

0.024

III

3.905 (2.621–5.818)

<0.001

4.165 (2.752–6.303)

<0.001

IV

1.335 (0.482–3.698)

0.578

1.081 (0.379–3.087)

0.884

Histology

Adenocarcinoma

0.835 (0.583–1.196)

0.325

0.859 (0.310–2.377)

0.769

Squamous cell

1.159 (0.802–1.673)

0.433

0.796 (0.265-2.397)

0.686

Biomarker

JAG1

1.025 (0.726–1.449)

0.887

1.037 (0.725–1.483)

0.842

CDCP1

1.456 (1.016–2.086)

0.041

1.282 (0.859–1.913)

0.225

FAP-α

1.313 (0.884–1.951)

0.177

1.489 (0.960–2.310)

0.076

Adjuvant Chemotherapy

0.829 (0.365–1.880)

0.653

0.830 (0.361–1.909)

0.661

Abbreviations ; JAG1: Jagged-1, CDCP1: CUB-domain-containing protein 1, FAP- α: Fibroblast activation protein-alpha

Table 3. Cox regression analysis for 5-year overall survival

Univariate analysis

Multivariate analysis

Parameter

HR (95 % CI)

P value

HR (95 % CI)

P value

Age (≥65)

2.086 (1.572–2.768)

<0.001

1.813 (1.274–2.578)

0.001

Male

1.724 (1.243–2.392)

0.001

1.542 (1.024–2.323)

0.038

Ever-smoker

1.732 (1.264–2.372)

0.001

0.934 (0.500–1.747)

0.832

Operation type

0.029

0.064

Lobectomy

1

1

Sublobar resection

1.888 (1.219–2.923)

0.004

0.946 (0.424–2.110)

0.892

Bilobectomy

1.151 (0.608–2.179)

0.667

0.435 (0.103-1.831)

0.256

Pneumonectomy

0.561 (0.139–2.261)

0.416

2.002 (1.148–3.490)

0.014

Stage

<0.001

<0.001

I

1

1

II

1.864 (1.281–2.712)

0.001

1.677 (1.006–2.795)

0.047

III

3.612 (2.592–5.031)

<0.001

4.961 (3.269–7.530)

<0.001

IV

3.556 (2.008–6.297)

<0.001

3.782 (1.919–7.455)

<0.001

Histology

Adenocarcinoma

0.624 (0.471–0.828)

0.001

0.809 (0.293–2.237)

0.683

Squamous cell

1.564 (1.175–2.082)

0.002

1.024 (0.342–3.068)

0.966

Biomarker

JAG1

1.121 (0.846–1.485)

0.427

1.315 (0.930–1.861)

0.122

CDCP1

1.498 (1.118–2.007)

0.007

1.170 (0.784–1.745)

0.443

FAP-α

1.092 (0.804–1.483)

0.573

1.103 (0.730–1.666)

0.640

Adjuvant Chemotherapy

0.966 (0.511–1.825)

0.915

0.855 (0.372–1.963)

0.712

Abbreviations ; JAG1: Jagged-1, CDCP1: CUB-domain-containing protein 1, FAP- α: Fibroblast activation protein-alpha

Table 4. Cox regression analysis for recurrence within 5 years after surgery in stage I non-small cell lung cancer

Univariate analysis

Multivariate analysis

Parameter

HR (95 % CI)

P value

HR (95 % CI)

P value

Age (≥65)

1.239 (0.711–2.162)

0.450

1.295 (0.737–2.273)

0.368

Male

1.654 (0.892–3.067)

0.110

1.449 (0.774–2.714)

0.247

Ever-smoker

1.490 (0.830–2.675)

0.182

0.864 (0.298–2.503)

0.788

Operation type

0.265

0.624

Lobectomy

1

1

Sublobar resection

1.799 (0.806–4.016)

0.152

1.194 (0.337–4.232)

0.784

Bilobectomy

1.731 (0.536–5.597)

0.359

1.516 (0.642–3.579)

0.342

Histology

Adenocarcinoma

0.580 (0.320–1.051)

0.073

0.913 (0.119–7.035)

0.930

Squamous cell

1.710 (0.934–3.132)

0.082

1.299 (0.632–2.670)

0.476

Biomarker

Jagged-1

1.443 (0.804–2.591)

0.219

1.353 (0.745–2.455)

0.321

CDCP-1

1.967 (1.104–3.504)

0.022

1.967 (1.104–3.504)

0.022

FAP-α

1.436 (0.736–2.805)

0.289

1.430 (0.729–2.807)

0.299

Adjuvant Chemotherapy

0.801 (0.195–3.295)

0.759

0.820 (0.194–3.463)

0.788

Abbreviations ; JAG1: Jagged-1, CDCP1: CUB-domain-containing protein 1, FAP- α: Fibroblast activation protein-alpha

Table S2: Risk factors for mortality in stage I patients assessed using Cox regression analysis

Univariate analysis

Multivariate analysis

Parameter

HR (95% CI)

P value

HR (95% CI)

P value

Age (≥65)

3.296 (2.036–5.336)

<0.001

3.420 (2.082–5.619)

<0.001

Male

1.781 (1.067–2.972)

0.027

1.302 (0.728–2.327)

0.373

Ever-smokers

1.663 (1.024–2.702)

0.040

1.067 (0.422–2.699)

0.890

Operation type

0.037

0.531

Lobectomy

1

1

Sublobar resection

2.224 (1.195–4.142)

0.012

1.230 (0.416–3.637)

0.708

Bilobectomy

1.485 (0.539–4.087)

0.445

1.448 (0.740–2.834)

0.279

Histology

Adenocarcinoma

0.462 (0.289–0.738)

0.001

1.874 (0.253–13.875)

0.539

Squamous cell

2.219 (1.384–3.557)

0.001

1.907 (1.181–3.081)

0.008

Biomarker

JAG1

1.132 (0.710–1.805)

0.604

1.228 (0.762–1.977)

0.398

CDCP1

1.595 (1.003–2.538)

0.049

1.530 (0.917–2.554)

0.104

FAP-α

1.064 (0.643–1.762)

0.808

1.103 (0.639–1.904)

0.725

Adjuvant Chemotherapy

0.243 (0.034–1.75)

0.160

0.239 (0.033–1.729)

0.156

Abbreviations ; JAG1: Jagged-1, CDCP1: CUB-domain-containing protein 1, FAP- α: Fibroblast activation protein-alpha
